# Comprehensive Phosphoproteomic Analysis of Pepper Fruit Development Provides Insight into Plant Signaling Transduction

**DOI:** 10.3390/ijms21061962

**Published:** 2020-03-13

**Authors:** Zhoubin Liu, Junheng Lv, Yuhua Liu, Jing Wang, Zhuqing Zhang, Wenchao Chen, Jingshuang Song, Bozhi Yang, Fangjun Tan, Xuexiao Zou, Lijun Ou

**Affiliations:** 1College of Horticulture, Hunan Agricultural University, Changsha 410128, China; hnliuzhoubin@163.com (Z.L.);; 2Longping Branch, Graduate School of Hunan University, Changsha 410125, China; junhenglv@hnu.edu.cn (J.L.); liuyuhua@hnu.edu.cn (Y.L.); wangjinghnu@hnu.edu.cn (J.W.); sjs122912@163.com (J.S.); 3Vegetable Institution of Hunan Academy of Agricultural Science, Changsha 410125, China; cszzq@126.com (Z.Z.); wencc1974@163.com (W.C.);

**Keywords:** abscisic acid, ethylene, fruit development, kinase, phosphorylation, phosphoproteome, pepper

## Abstract

Limited knowledge is available for phosphorylation modifications in pepper (*Capsicum annuum* L.), especially in pepper fruit development. In this study, we conducted the first comprehensive phosphoproteomic analysis of pepper fruit at four development stage by Tandem Mass Tag proteomic approaches. A total of 2639 unique phosphopeptides spanning 1566 proteins with 4150 nonredundant sites of phosphorylation were identified, among which 2327 peptides in 1413 proteins were accurately quantified at four different stages. Mature Green (MG) to breaker stage showed the largest number of differentially expressed phosphoproteins and the number of downregulated phosphoproteins was significantly higher than that of upregulated after MG stage. Twenty seven phosphorylation motifs, including 22 pSer motifs and five pThr motifs and 85 kinase including 28 serine/threonine kinases, 14 receptor protein kinases, six mitogen-activated protein kinases, seven calcium-dependent protein kinases, two casein kinases, and some other kinases were quantified. Then the dynamic changes of phosphorylated proteins in ethylene and abscisic acid signaling transduction pathways during fruit development were analyzed. Our results provide a cascade of phosphoproteins and a regulatory network of phosphorylation signals, which help to further understand the mechanism of phosphorylation in pepper fruit development.

## 1. Introduction

Pepper (*Capsicum annuum* L.) is an important vegetable crop of solanaceae plants in China, which is also widely planted and the largest seasoning crop in the world [1]. Capsaicin, flavor, pigment (anthocyanins and carotenoids), vitamin C, vitamin E, and other vitamins can be synthesized and accumulated in capsaicin fruits [2], which are considered as good sources of the most basic nutrients and has important economic value [3]. However, fruit development is a complex process of gene regulation, including the absorption, accumulation, and metabolism of soluble solids, sugar acids, vitamin C, and some specific substances, involving sensory changes in texture such as aroma, smell, appearance, and color [4]. Recently, numerous studies have focused on transcriptome and proteome profiles to reveal pepper fruit ripening mechanisms [1,5,6]. However, the central principle and recent functional studies of proteins have revealed that there are many complex regulatory processes from genes to proteins, such as transcription, translation, and post-translational modification. The precursor proteins need to undergo a series of post-translational modifications (PTM) to form functionally active proteins. PTM plays an important role in changing the structure of precursor protein, enzyme activity, substrate specificity, mediating the interaction between protein and nucleic acid, lipid, protein, and other molecules, regulating protein subcellular localization, protein complex formation, and protein degradation [7].

In vivo, phosphorylation is the most extensive covalent modification in PTM of proteins and the most important regulatory modification in prokaryotes and eukaryotes [8]. Phosphorylation plays an important role in regulating the normal function of proteins, and studies have shown that the activation of protein kinases, development, variation, and transformation of cells were all regulated by protein phosphorylation [9,10,11]. Moreover, reversible phosphorylation of protein is the most important way of signal transmission in organisms [12]. One third of the proteins in eukaryotes are phosphorylated at any time [13], and the genes encoding protein kinase and phosphorylase account for 2–4% of the whole genome [14]. Therefore, understanding the regulatory mechanism of protein phosphorylation during development is of great significance to the development and quality formation of pepper fruit.

Protein phosphorylation in cells is a dynamic process, and subtle differences may lead to changes in the level of cell metabolism. Therefore, the influence of protein phosphorylation on plant growth and development is omnidirectional. In recent years, many large-scale phosphoproteomic analyses in different plant species were performed [15,16,17,18]. Agrawal et al. [19] systematically quantified and analyzed the phosphorylated proteins of developing rapeseed for the first time, and revealed that more than 44% of the enzymes identified in the phosphorylated proteins participate in various metabolic pathways. Chitteti et al. [20] analyzed 53 putative phosphorylated proteins during the study of Arabidopsis cell dedifferentiation, and nine of them were differentially expressed during cell dedifferentiation. Chao et al. [21] found that under light induction, the photophosphorylpyruvate carboxylase (PEPCK) of *Arabidopsis thaliana* was phosphorylated at Ser55, Thr58, and Thr59, the degree of phosphorylation at Ser55 was negatively correlated with PEPCK activity, indicating that the phosphorylation of Ser55 could inhibit the light-induced decarboxylation of PEPCK. Medzihradszky et al. [22] found that if the Phytochrome B at Ser86 in arabidopsis leaves was phosphorylated, the activity of Phytochrome B would be inhibited, thus hinding the plant’s light-induced signal transduction.

Recently, studies on protein phosphorylation in solanaceous crops have been reported. Anguenot et al. [23] found that the mechanism of sucrose synthase partition in tomato fruit was regulated by calcium and protein phosphorylation/dephosphorylation. Stulemeijer et al. [24] using a label-free approach identified 50 phosphoproteins, found some new phosphoproteins changes in abundance during the very early stages of hypersensitive response development in tomato seedlings, suggesting that photosynthetic activity is specifically suppressed in a phosphorylation-dependent way. Elagamey et al. [25] identified 35 phosphoproteins by mass spectrometry of potato extracellular matrix, found these phosphoproteins mainly control cell wall integrity, metal flux, and redox homeostasis. However, limited knowledge is available for phosphorylation modifications in pepper, especially in pepper fruit development. 

Therefore, on the basis of our previous study of the transcriptome and proteome of pepper fruit development [26], we further conducted a comprehensive phosphoproteomic analysis of one pepper variety SJ11-3 fruits at four different developmental stages for the first time. In this study, we identified a total of 4150 phosphorylation sites in 2639 phosphorylated peptides from 1566 phosphoproteins, revealed a complicated phosphorylation signal transduction network during fruit development and ripening, and provide fresh insights into the global protein phosphorylation events in pepper fruit development. 

## 2. Results and Discussion

### 2.1. Phosphoproteomic Profiling

In an effort to explore the roles of protein phosphorylation in pepper fruit development, we profiled the phosphoproteome of four different developmental stages by using quantitative Tandem Mass Tag (TMT) phosphoproteomic approach. A total of 2639 unique phosphopeptides spanning 1566 proteins with 4150 sites of phosphorylation were identified (Figure 1A), among which 2327 peptides in 1413 proteins were accurately quantified (Appendix A). Among the amino acids that can be phosphorylated, serine, threonine, and tyrosine are the main three amino acids that affect protein function [27]. Of the 4150 nonredundant phosphorylation sites, the quantities of phosphoserine (pSer), phosphothreonine (pThr) and phosphotyrosine (pTyr) are 3423 (82.48%), 683 (16.46%), and 44 (1.06%), respectively (Figure 1B). Phosphoproteomic analyses in other plants, such as Arabidopsis [28,29], maize [30], and wheat [31], found a similar distribution of phosphorylated residues and that Ser phosphorylation sites accounted for more than 80% while Tyr phosphorylation sites accounted for no more than 2%. A comparison of the phosphopeptides and their corresponding phosphoproteins identified from the four different stages pepper showed that 2312 phosphopeptides and 1406 phosphoproteins were identified in both four different stages, indicating that most phosphopeptides and phosphoproteins appear at all stages of development (Appendix A).

### 2.2. Characteristics of the Phosphorylation Sites

Among the 1566 phosphoproteins, 652 (41.63%) with only one phosphorylated site, 391 (24.97%) with two phosphorylated sites, and 523 (33.40%) with three or more phosphorylated sites (Figure 1C and Appendix A), and the average distribution of modification sites per 100 amino acids was 2.44 (Figure 1D). Of all proteins with more than three phosphorylation sites, 38 had more than 10 phosphorylation sites, and, in particular, six had more than 15 phosphorylation sites. There were 41 phosphorylation sites (36Ser + 5Thr) of nucleolin 1 (A0A1U8FF91) protein, hepatoma-derived growth factor-related protein 2 (A0A1U8GUL7) has 27 phosphorylation sites (23Ser + 4Thr), and serine/arginine-rich splicing factor SR45 isoform X1 (A0A1U8EN90) has 25 phosphorylation sites (22Ser + 3Thr). Among all phosphoproteins, the phosphrylation of nucleolin 1 (A0A1U8FF91) occurs most in Ser sites, which was identified in 36 Ser phosphorylation sites. pre-mRNA-splicing factor CWC22 homolog isoform X1 (A0A1U8GL40) and splicing factor 3B subunit 1 (A0A1U8EIB4) have seven Thr phosphorylation sites, respectively. However, Hexosyltransferase (K4B482) and Uncharacterized protein (K4CJ72) showed the largest phosphorylation of Tyr sites, each with three Tyr phosphorylation sites.

### 2.3. Identification of Differentially Expressed Phosphoproteins (DEPPs) at Different Stages

During the development of pepper fruit, there was a significant difference in the number of differentially expressed phosphoproteins (DEPPs) between adjacent development stages, among which Breaking/Mature Green (Br/MG) had the most DEPPs, while Mature Green/ Immature Green (MG/IMG) had the least (Table 1 and Appendix A). This indicated that the change of pepper fruit was the biggest from MG stage to Br stage, which led to the significant changes of the expression of phosphoprotein in the fruit. In the three adjacent development stages, the DEPPs were mainly upregulated in MG/IMG, and the upregulated protein accounted for 74.75% of the total DEPPs, while the DEPPs in Br/MG and Mature Red/Breaking (MR/Br) were mainly downregulated, which accounted for 97.45% and 98.35% of the total DEPPs, respectively. This may indicate that before the MG stage, the expression level of some phosphoproteins in the fruit increased significantly to promote the growth and development, but after the fruit grew to the MG stage, the expression abundance of most phosphoproteins in the fruit decreased significantly to promote the ripening of the fruit.

Further analysis of differentially expressed phosphorylated proteins revealed that 69 phosphoproteins showed significant changes in all three adjacent comparison groups, while 3, 314, and 86 phosphoproteins showed significant changes only in MG/IMG, Br/MG, and MR/Br, respectively (Figure 2A). Among the upregulated phosphoproteins, there were no co-up-regulated phosphoproteins in the three comparison groups, but the phosphoproteins of ABSCISIC ACID-INSENSITIVE 5-like protein 7 (A0A1U8E6Y8), protein SRC2-like (A0A1U8GWU7), SNF1-related kinase (A1IKU3), and Uncharacterized protein (M0ZTY3) were significantly increased in the MG/IMG and Br/MG comparison groups (Figure 2B). ABSCISIC ACID-INSENSITIVE 5-like protein 7 and SNF1-related kinase are closely related to the abscisic acid (ABA) signal transduction pathway, and this suggests that changes in ABA signaling pathway may play an important regulatory role in the growth of capsicum fruit. Among the downregulated phosphoproteins, 12 phosphoproteins including Replication factor C subunit 1 (A0A1U8H887), serine/arginine-rich splicing factor SR30-like (A0A1U8GY34), and Eukaryotic translation initiation factor 3 subunit C (A0A1U8DXK3) were significantly decreased in the three adjacent comparison groups (Figure 2C). Replication factor C is an important factor involved in DNA replication and repair mechanisms as well as cell proliferation [32,33]. Liu et al. [34] found that AtRFC1 plays an important role in the repair of double-stranded fractures when homologous chromosomes are recombined during meiosis. Serine/arginine-rich splicing factor is a key regulatory protein that controls the occurrence of alternative splicing. It is widely involved in RNA processing, including RNA splicing, mRNA nucleation, mRNA stability, and translation [35]. Phosphorylation can increase the role of SR protein in the splicing mechanism of precursor mRNA, while reversible phosphorylation is beneficial to the assembly of spliceosomes [36]. Lopato et al. [37] showed that if the atsR3P0 protein level of transgenic arabidopsis thaliana increased, the transition from nutrition to reproductive period was delayed, the life cycle was prolonged, and the individual size of transgenic plants increased. Eukaryotic translation initiation factor 3 subunit C complex can play an important role in the formation of pre-translation initiation complex, it can be combined with different eIF and RNA to regulate the whole process of translation initiation, and selectively regulate the synthesis of proteins through the translation initiation of different types of mRNA, so as to regulate the growth of cells [38]. These proteins play important roles in the regulation of cell development, cell differentiation, and cell cycle. Therefore, in the early stage, the abundance of these phosphoproteins is relatively high, which leads to the acceleration of cell development and differentiation in the fruit, the shortening of cell cycle, which is conducive to the expansion and growth of the fruit. With the continuous development of the fruit, the abundance of these phosphoproteins continues to decrease, the development and differentiation of cells slows down, and the fruit gradually matured.

### 2.4. Motif Analysis of Lysine Phosphorylated Peptides

To identify the possible specific motifs flanking phosphorylated lysine, MEME Software [39] were used in this study to extract the over-represented motifs of amino acids. A total of 27 phosphorylation motifs, including 22 pSer motifs and five pThr motifs, were defined on 3322 unique peptides (Figure 3A, Appendix A). Ma et al. [11] defined 17 pSer and two pThr in the phosphoproteomic of cotton mutant fiber development, but also did not define pTyr. These motifs exhibited different abundances, in 22 pSer motifs, motifs [sP] and [Rxxs] occupied the highest proportion of 450 and 269 peptides, respectively. As reported by many studies, [sP] and [Rxxs] were frequently recurring motifs [40,41], [sP] motifs were potential substrates of MAPK, cyclin-dependent kinase (CDKs), and CDK-like kinase, and [Rxxs] were recognized by calcium-/calmodulin-dependent PK II [18]. While, the phosphorylated peptides with the lowest proportion in these motifs were [sDDE], which only account for 50 of all identified peptides. In the four pThr motifs, motifs [tP] and [txP] occupied the highest and the lowest proportion of 180 and 51 identified phosphorylated peptides, respectively (Figure 3B). It indicated that the Serine/Threonine residues around some neutral and alkaline residues (P and R) may be more easily phosphorylated. 

In our study, according to the heat map of the amino acid compositions surrounding the phosphorylation sites, proline (P) and arginine (R) were significantly over-represented in positions +1 and −3, respectively, other residues like leucine (L), methionine (M), and tyrosine (Y) were highly present in position −5, cystine (C), phenylalanine (F), glutamic acid (E), and tryptophan (W) were highly present in positions −6, +1, +3, and +4, respectively. While, the occurrence of histidine (H), lysine (K) in position +1, and tryptophan (W) and tyrosine (Y) in position +2 were lowest (Figure 3C).

### 2.5. Secondary Structure and Subcellular Localization Analysis

Next we used the algorithm NetSurfP [42] to analyze the surface accessibility and second structural features of phosphorylation sites in proteins. In our study, results showed that the pSer, pThr, and pTyr showed different preferences for secondary structure compared with those that all identified Ser/Thr/Tyr (Figure 4A–C). In phosphorylated sites, the possibility over 90% of pSer and pThr were distributed in regions predicted to be disordered (Coil). A similar distribution was obtained from rice, implying a preference for coils with respect to the phosphorylation of both serine and Threonine [43]. In addition, the p-Ser/Thr/Tyr that are located in the ordered region are more likely to be found in alpha-helix than in beta-strand. Compared with those that all identified Ser/Thr/Tyr, the possibility of pSer, pThr, and pTyr occurred more frequently in unstructured regions while the opposite was shown in ordered regions. Surface accessibility of the phosphorylation sites was also analyzed, and 51.4%, 45.1%, and 29.7% of pSer, pThr, and pTyr, respectively, were located on the protein surface, compared with 65.1%, 60.7%, and 33.3% of all Ser, Thr, and Tyr, respectively (Figure 4A–C). The results showed the surface accessibility of all Ser/Thr/Tyr are higher than those of p-Ser/Thr/Tyr, indicating phosphorylation is likely to slightly affect the surface property of modified proteins, resulting in a decrease in their surface properties in pepper. However, Chen et al. [44] suggested that phosphorylation could lead to increased surface properties of Phaeodactylum tricornutum. Further research is needed to determine what causes the decrease in surface property at phosphorylation sites in peppers.

The CELLO [45] algorithm analysis showed that 647, 209, 117, and 63 of the phosphorylation proteins were located in the nucleus, cytoplasm, plasma membrane, and chloroplast, respectively. In contrast, phosphorylation proteins in the remaining compartments such as mitochondrial, extracellular accounted for only 5.82% in total (Figure 4D). A previous study indicates that protein phosphorylation occurs in all subcellular compartments, and more than 40% of phosphoproteins occurred in the nucleus [29,46], and this is consistent with our results. In our study, in addition to the nucleus, the organelle with the most protein localization is the cytoplast, which may be related to the fact that the proteins containing motif [SP] are basically located in the nucleus and cytoplasm [40]. However, it was found that the chloroplast was the most localized organelle except for the nucleus in rice [43]. We speculate that this difference may be due to the different organ parts of the sample. The samples we studied were fruit, with fewer chloroplast organelles compared with leaves and less use of chloroplast function in the late stage of fruit development, resulting in less protein localization in the chloroplast.

### 2.6. Top 20 Most Abundant GO Term Analysis of DEPPs

The top 20 most abundant GO terms were selected for further analysis, and the number of terms for biological processes, molecular functions, and cell components was nine, four, and seven, respectively (Figure 5). Cellular process and metabolic process were predominant in the category of biological processes, and the numbers of these two categories were significantly higher than that of the other category. In the molecular function, the largest group was binding and catalytic activity. The cellular components of these proteins mainly included cell and cell parts. In the three comparison groups, the Br/MG had more DEPPs than MR/Br and MG/IMG. This indicates that during fruit development, the number of DEPPs at Br/MG stage is the largest, which may be directly related to our earlier study, finding that the number of differentially expressed proteins at Br/MG stage is much higher than that of other comparison groups [26]. It also found that the binding, catalytic activity, and cellular process were highly involved in the development of pepper.

### 2.7. Top 20 Most Abundant KEGG Analysis of DEPPs

A top 20 most abundant KEGG pathway analysis was also performed to reveal the phosphoprotein-associated pathways in the process of pepper development (Figure 6). In the three comparison groups, the DEPPs counted in MG/IMG, Br/MG, and MR/Br were 39, 195, and 127, respectively. These phosphoproteins are mainly distributed in splicesome, RNA transport, protein processing in the endoplasmic reticulum, plant–pathogen interaction, mRNA surveillance pathway, and PI3K-Akt signaling pathway. We found that the total number of DEPPs identified in the three comparison groups were significantly higher than the number of phosphoprotein species identified throughout the development stages, and this is because some phosphoproteins may change significantly not only in one comparison group, but also in multiple adjacent comparison groups, resulting in multiple counts. Especially in the spliceosome pathway, a total of 55 phosphoproteins were counted in the three comparison groups, while only 33 phosphoprotein species were identified in the whole development stage. It suggested that there may be a large number of DEPPs in the spliceosome pathway that underwent significant changes at multiple developmental stages. During the development of anthers and pollens of male sterility kenaf cytoplastmic lines, large amounts of differentially phosphorylated proteins were also found in the splicesome pathway [47]. Zeng et al. [48] found that during the development of citrus fruits, almost an entire set of proteins involved in the glycolysis pathway in chromoplast were phosphorylated. However, the number of DEPPs in the glycolysis pathway was small in this study, and this may be related to differences in the nutritional composition of different fruits. We also found that in the top 20 KEGG pathways, the number of DEPPs in Br/MG was significantly higher than that in MG/IMG and MR/Br, but in the spliceosome, mRNA surveillance pathway, and MAPK signaling pathway–plant pathways, the number of DEPPs in Br/MG was 26, 12, and 8, respectively, while that in MR/Br was 23, 8, and 6, respectively. In these three pathways, the DEPPs in the MR/Br decreased only slightly by 11.54%, 33.33%, and 25%, respectively, compared with those in the Br/MG. This indicates that spliceosome, mRNA surveillance pathway, and MAPK signaling pathway–plant pathways, may play an important regulatory role in the ripening process from the color change stages (MG to MR stage).

### 2.8. The 120 Kinase Analysis in Phosphorylated Proteome

Protein kinases are intracellular message-dependent enzymes that mediate and amplify protein phosphorylation and assist in signaling. These enzymes use ATP or GTP as donors of phosphate groups and transfer phosphate groups to specific amino acid residues in their substrates. Lv et al. [18] using large-scale phosphoproteomic profiling identified 85 protein kinase in two seedling leaves of bread wheat. In this study, a total of 268 phosphopeptides spanning 120 kinase were identified, among which 140 peptides in 85 kinase were accurately quantified, and the number of phosphoserine, phosphothreonine, and phosphotyrosine was 112 (80%), 22 (15.7%), and 6 (4.2%), respectively (Appendix A). These quantitative kinase proteins include 28 serine/threonine kinases (PLKs), 14 receptor protein kinases (RPKs), six mitogen-activated protein kinases (MAPKs), seven calcium-dependent protein kinases (CDPKs), two casein kinases (CKs), and some other kinases. MAPKs in eukaryotes play a pivotal role in transferring exogenous stimulation signals to the downstream of the intracellular response receptor/sensor [49]. When stimulated by the extracellular environment, MAPKKK activates MAPKK through phosphorylation of the serine-threonine site, and then MAPKK phosphorylates serine and tyrosine sites of the T-x-Y motif in the MAPK protein active loop. Active MAPK enables other downstream protein kinases, metabolites, transcription factors, and cytoskeleton components to perform cell functions such as cell division and differentiation [50,51]. Further analysis showed that our research includes seven MAPKs (A0A0V0I4E2, A0A0V0IXF4, A0A1U8F3L5, and M1CT49), one MAPKK (A0A088BHA3), and one MAPKKK (A0A1U8EWT2). In arabidopsis thaliana, MPK4 phosphorylates MAP65-3/PLE protein and participates in cell membrane separation and cell plate formation, thus completing cytokinesis [52]. Zhou et al. [53] found that knockout of the *MKK9* or *MKK6* gene could delay the senescence of Arabidopsis leaves, while in *MPK6* mutants, overexpression of MKK9 resulted in a decrease in the degree of premature senescence. These results indicate that the MAPK signal cascade of MKK9-MPK6 also plays an important role in the regulation of leaf senescence. The differential expression of MAPKs in different tissues and organs indicates that MAPKs are involved in different developmental processes of plants [54,55]. In our study, the phosphorylation level of M1CT49 in several MAPKs showed no significant change, while A0A0V0I4E2 and A0A0V0IXF4 significantly decreased in the Br stage, and A0A1U8F3L5 significantly decreased in the MR stage. The phosphorylation level of MAPKK (A0A088BHA3) increased significantly in the MG stage and decreased significantly in the Br stage. The phosphorylation level of MAPKKK (A0A1U8EWT2) decreased significantly during the Br stage. The variation of phosphorylation level of different MAPKs at different stages were different, suggesting that MAPKs may be involved in the whole development process of pepper fruit, while different MAPKs may play different roles.

### 2.9. Plant Hormone Signal Transduction in Phosphorylated Proteome

Protein phosphorylation/dephosphorylation mediated by protein kinase/phosphatase is critical to ethylene signaling [56]. ETR and CTR1 protein, as negative regulators of the ethylene signaling pathway in endoplasmic reticulum, had no significant change in the phosphorylation level during the whole process of pepper fruit development (Figure 7), indicating that the phosphorylation level of these two proteins was not affected by the change of ethylene content during the process of pepper fruit development, but functioned normally. Based on the similarity of Ctr1 and RAF protein kinase in sequence, it has been speculated that CTR1, as a MAPKKK, can activate the downstream ethylene signal by mediating a MAPK cascade reaction in the ethylene signaling pathway [57,58]. EIN2 is the first positive regulator of ethylene response [59], and its functionally deficient mutants exhibit a strong ethylene complete insensitivity phenotype in arabidopsis thaliana [60]. ERFs is a transcription factor located downstream of EIN3/EIL1, EIN3 can bind to the cis-acting factor PERE site of the ERF1 promoter to induce the gene expression, and ERF1 can induce gene expression by binding to the GCC box of many secondary ethylene reaction gene promoters, thus causing an ethylene reaction [61]. When intracellular ethylene concentration is insufficient, the ETR receptor activates the CTR1 protein, and the activated CTR1 can inactivate it by directly phosphorylating the C-terminal of EIN2 [62], inhibiting the downward transmission of the ethylene signal and thus limiting the ethylene response. Therefore, when ethylene content was low in the early stage of fruit development (IMG and MG), the phosphorylation level of EIN2 (A0A1U8E8G2) protein was relatively high, and the phosphorylation level of MPK6 (A0A0V0I4E2) upstream of EIN2 and ERF1 (M1BG39) downstream of EIN2 were also relatively high. However, when the concentration of ethylene in cells is high, the receptor binds to the hormone and becomes inactive, then turns off the CTR1 protein in turn. The deactivated CTR1-coding protein blocks the phosphorylation of the positive regulator EIN2 through a series of phosphorylation cascaded reactions, resulting in a decrease in the phosphorylation level of EIN2. Chen et al. [63] showed that in the presence of ethylene, EIN2 lacks phosphorylation on multiple serine and threonine residues. The C-terminal of EIN2 was splintered by an unknown mechanism and moved to the nucleus, where EIN3/EIL1 were stabilized and ERF1/2 degradation was induced. Thus, the *EIN2* and *EIN3* genes encode proteins and induce ethylene reaction [64]. Therefore, when ethylene content in fruits increased in the Br and MR stages, even though MPK6, EIN2, and ERF1 proteins in the signaling chain had two, six, and four phosphorylation sites, respectively, but their phosphorylation level in the Br stage was significantly lower than that in the MG stage. This suggests that the ethylene signal transduction during the development of pepper fruit may involve the phosphorylation cascade reaction, and the phosphorylation plays a key role in the ethylene signal transduction. At the same time, we found that during the whole process of fruit development, the phosphorylation level of EIN3, an important transcription factor downstream of EIN2 regulating ethylene reaction, did not change significantly. It indicates that the transmission of the ethylene signal may be mainly affected by the changes in the phosphorylation levels of MPK6, EIN2, and ERF1 proteins. 

Numerous studies have shown that phosphorylation/dephosphorylation also plays an important role in ABA signal transduction [65,66]. Wang et al. [67] and Umezawa et al. [68] have applied a phosphoproteomic approach to an *Arabidopsis* SnRK2-disruptant (srk2dei), and successfully detected a number of phosphopeptides with differential phosphorylation between the wild type (WT) and srk2dei. Studies suggested that ABA signaling changes during after-ripening are more critical than content changes for the regulation of dormancy and germination in barley grains [69]. The core ABA signaling pathway includes ABA-specific receptors (PYR1/PYLs/RCARs) and their downstream phosphatases (clade-a PP2C) and protein kinases (SnRK2). During fruit development, ABA signaling is activated by receptor proteins such as PYRs and transmitted to downstream transcription factors and cis-elements through reversible phosphorylation of ABI1 and SnRK2 proteins [70], ultimately triggering the expression of ripening related genes such as fruit softening, sugar accumulation, and coloring. In rice seed development, 13 PP2Cs and three SnRK2s showed differentially phosphorylated patterns, and the majority of differentially phosphorylated ABA-related proteins are functionally related to ABA signaling [71]. Studies on the ABA receptor showed that downregulation of *FaPYR1* expression in strawberry fruits could inhibit fruit ripening [72,73]. However, the downregulation of 2C protein phosphatase gene *FaABI1* promoted the ripening of strawberry fruits [74], indicating that *FaPYR1* was involved in the ripening of strawberry fruits as a positive regulatory factor, while *FaABI1* was involved in the ripening of strawberry fruits as a negative regulatory factor. In our study, there was no significant change in the phosphorylation level of PYR/PYL, ABI1 (A0A0K0NLC3) phosphorylation was at a high level in the early stage (IMG and MG stages), and then decreased in Br and MR stages. Two proteins with serine/threonine kinase (SNF1-related kinase (A1IKU3) and SRK2B (A0A1U8E5G8) functions were identified in our study. The phosphorylation trend of SRK2B was basically consistent with ABI1, while the phosphorylation level of SNF1-related kinase (A1IKU3) increased significantly from IMG to Br, and decreased significantly in the MR stage. However, as the phosphorylation level of SNF1-related kinase at all stages was significantly higher than that of SRK2B, we believe that the phosphorylation level of SnRK2 in our study was mainly influenced by the SNF1-related kinase. Research showed that PYR1/PYL/RCARs could not bind to phosphatase PP2Cs without ABA, so the activity of PP2C was very high, which prevented the activation of kinase SnRK2. In the presence of ABA, PYR1/PYL/RCARs bind and inhibit the activity of PP2Cs, which makes the phosphorylated SnRK2 accumulate, and then phosphorylates the downstream ABA response binding factor ABFs [75]. Therefore, when ABA content was low in the early stage, ABI1 phosphorylation level was high, which prevented the activation of SnRK2-related proteins and led to a low level of phosphorylation of SNF1-related kinase, which hindered the downward transmission of ABA signal. At the same time, ABI5 (v5l071) downstream of the signaling chain was at a high level of phosphorylation in the early stage. ABI5 could encode transcription factors that function in the last step in the ABA signaling pathway [76]. Increased ABI5 expression is sufficient for hypersensitivity to inhibition of growth by ABA or sugar [77], and loss-of-function mutants of ABI5 could display insensitivity to ABA in seedling development [78], and this further reduced the sensitivity of fruit to ABA. The inhibition of ABA signal in the early stage and the decrease of sensitivity to ABA can effectively avoid the phenomenon of falling flowers and fruits caused by excessive sensitivity to ABA. However, when ABA content gradually increased in the Br stage, ABI1 activity was inhibited, and the phosphorylation level was significantly lower than that in the MG stage, while the phosphorylation level of SNF1-related kinase was increased, leading to the accumulation of phosphorylated SnRK2 protein. As for ABI1, the phosphorylation level was lower in the MR stage, and the phosphorylation level of SNF1-related kinase was significantly lower than that of the Br stage, but the fruit was still maturing rapidly. This may be due to the fact that the phosphorylation degree of ABI5 downstream is consistent with the Br stage, which is at a low level, leading to the increased sensitivity of pepper fruits to ABA, which promotes fruit maturation. 

## 3. Materials and Methods 

### 3.1. Plant Growth and Sampling

One pepper (*Capsicum annuum* L.) variety, ‘SJ11-3′ was provided by Hunan Vegetable Research Institute (Changsha, China), and grown in a greenhouse (16 h of light at 30 ± 2 °C and 8 h of darkness at 20 ± 2 °C). Fruits were collected at 20 days after anthesis (DAA) (immature green, IMG), 30 DAA (mature green, MG), 40 DAA (breaker, Br), and 50 DAA (mature red, MR). Fruit pericarps at each developmental stage were collected and pooled from three individual plants. The fruit pericarp tissues were frozen in liquid nitrogen, stored at −80 °C for phosphoproteome analyses. Three biological replicates for each time point were performed.

### 3.2. Protein Extraction and FASP Digestion

Samples were added to SDT Buffer to extract the protein [79]. The filtrate was quantified with the BCA Protein Assay Kit (Bio-Rad, Berkeley, CA, USA) and then stored at −80 °C. Around 20 µg of proteins were taken from each sample and trypsin hydrolysis was performed by FASP method [79]. The peptides were desalted on C18 Cartridges (Empore™ SPE Cartridges C18 (standard density), bed I.D. 7 mm, volume 3 mL, Sigma, St. Louis, MO, USA), and then lyophilized and dissolved in 40 µL of 0.1% (*v*/*v*) formic acid. The peptide content was estimated by UV light spectral density at 280 nm.

### 3.3. TMT Labeling and Enrichment of Phosphorylated Peptides by the TiO_2_ Beads

About 600 μg peptide mixture of each sample was labeled using TMT reagent [80] according to the manufacturer’s instructions (Thermo Fisher Scientific, Waltham, MA, USA). The lyophilized samples of the FASP digested peptides were redissolved in 500 µL 1×DHB buffer. Then, TiO_2_ beads were added and agitated for 2 h. The centrifugation was carried out for 1 min at 5000× *g*, resulting in the beads. Additionally, they were washed with 50 µL of washing buffer I three times and then 50 µL of washing buffer II three times. Finally, the phosphopeptides were eluted with 50 µL of elution buffer three times, followed by lyophilization and MS analysis.

### 3.4. HPLC and LC-MS/MS Analysis

Each fraction was injected for nanoLC-MS/MS analysis. The peptide mixture was loaded onto a reverse phase trap column (Thermo Scientific Acclaim PepMap100, 100 μm × 2 cm, nanoViper C18, Thermo Fisher Scientific), which was connected to the C18-reversed phase analytical column (Thermo Scientific Easy Column, 10 cm, ID75 μm, 3μm, Thermo Fisher Scientific) in buffer A (0.1% Formic acid) and separated by linear gradient buffer B (84% acetonitrile and 0.1% Formic acid) at a flow rate of 300 nL/min and controlled by IntelliFlow. A 4-h linear gradient was then performed: 0–55% buffer B for 220 min, 55–100% buffer B for 8 min, hold in 100% buffer B for 12 min.

LC-MS/MS analysis was performed on a Q Exactive HF-X mass spectrometer (Thermo Scientific) that was coupled with Easy nLC (Thermo Fisher Scientific) for 240 min. The mass spectrometer was operated in the positive ion mode. A data-dependent top10 method was used to acquire MS data, which dynamically selected the most abundant precursor ions from the survey scan (300–1800 m/z) for HCD (higher energy collisional dissociation) fragmentation. Automatic gain control (AGC) target was set to 3e6, maximum inject time to 50 ms, and dynamic exclusion duration to 60 s. Survey scans were acquired at a resolution of 70,000 at m/z 200 and resolution for HCD spectra was set to 17,500 at m/z 200, and isolation width was 2 m/z. Normalized collision energy was 30 eV and the underfill ratio was defined as 0.1%. The instrument was run with peptide recognition mode enabled.

### 3.5. Data Analysis

MS/MS spectra were searched using MaxQuant software version1.5.3.17 (Max Planck Institute of Biochemistry in Martinsried, Germany) [81]. The search was performed using the uniprot_solanoideae_178676_20170905 database (http://www/uniprot.org/uniprot/?query=Capsicum+annuum&sort=score). Phosphoproteins were identified with the following parameters: trypsin enzyme with the number of missed cleavages up to two, carbamidomethyl as the fixed modification, the fixed modification was carbamido methylation for cysteines, TMT 6plex (K), TMT 6plex (N-term), the variable modifications were phosphorylation modification for lysines, tyrosine and serine. The false discovery rate (FDR) threshold for modification site was 0.01, the minimum peptide length was five, and a MaxQuant score was set at ≥20. The charge states of peptides were +1, +2, and +3. Peptide mass tolerances were set at ±20 ppm for all MS1 spectra acquired, and fragment mass tolerances were set at 0.1 Da for all MS2 spectra acquired. Phosphoproteins identified in at least two of the three replicates were considered for expression analysis. The protein ratios are calculated as the median of only unique peptides of the protein. All peptide ratios were normalized by the median protein ratio. The median protein ratio should be 1 after the normalization. For LFQ, a fold change of ≥1.2 or ≤0.8 was regarded as significantly up- or downregulated, respectively.

### 3.6. Gene Ontology (GO) and KEGG Annotation

Sequences of phosphoproteins were retrieved from the UniProtKB database (Release 2016_10). The retrieved sequences were locally searched against the NR database using the NCBI BLAST+ and InterProScan [82] to find homologous sequences from which the functional annotation can be transferred to the studied sequences. Blast2GO [83] was then used to annotate sequences and map gene ontology (GO) [84] (Version 3.3.5, BioBam, Valencia, Spain). The GO annotation results were plotted by R scripts. Then the sequences of phosphoproteins were blasted against the online Kyoto Encyclopedia of Genes and Genomes (KEGG) database (http://geneontology.org/) to retrieve their KOs and were subsequently mapped to pathways in KEGG [85]. The corresponding KEGG pathways were extracted.

### 3.7. Motif Prediction

The sequence information of 13 amino acids (including the modification sites and ±6 amino acids of the modified sites) were extracted to predict the possible conservative motifs by using the MEME (http://meme-suite.org/index.htm).

### 3.8. Accession Numbers

The mass spectrometry proteomics data is available from the ProteomeXchange Consortium (https://www.ebi.ac.uk/pride/archive/) via the PRIDE partner repository with the dataset identifier PXD017648. Username: reviewer81895@ebi.ac.uk, Password: t6n96DVf. 

## 4. Conclusions

To understand the phosphosignaling that takes place during development of pepper fruit, phosphoproteomic profiles were obtained from pepper at four different stages. In this study we identified 1566 phosphoproteins, among which 1413 phosphoproteins were accurately quantified. Results showed that 85 kinase, such as PLKs, RPKs, MAPKs, CDPKs, and CKs, were involved in pepper fruit development. Plant signaling transduction pathway analysis found that the transmission of ethylene signal may be mainly affected by the changes in the phosphorylation levels of MPK6, EIN2, and ERF1 proteins. The phosphorylation levels of these proteins increase in the early stage of development to limit the transmission of ethylene signal, while the phosphorylation levels decrease in the late stage of development to promote the transmission of ethylene signal and accelerate the fruit ripening. The ABA signal transduction was mainly influenced by the changes in the phosphorylation levels of ABI1, SNF1-related kinase, and ABI5 during the fruit development of pepper, among which ABI5 downstream of the transmission chain may play a more important regulatory role in ABA signal transmission.

## Figures and Tables

**Figure 1 ijms-21-01962-f001:**
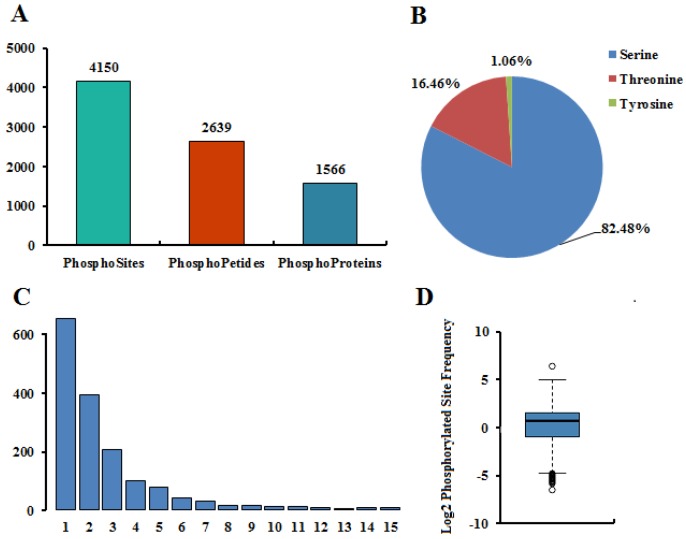
Summary of phosphoproteomic data. (**A**) Statistical results of phosphoproteome identification. (**B**) Distribution of hosphorylation sites. (**C**) The number of hosphorylation sites in protein. (**D**) Frequency distribution of hosphorylation sites.

**Figure 2 ijms-21-01962-f002:**
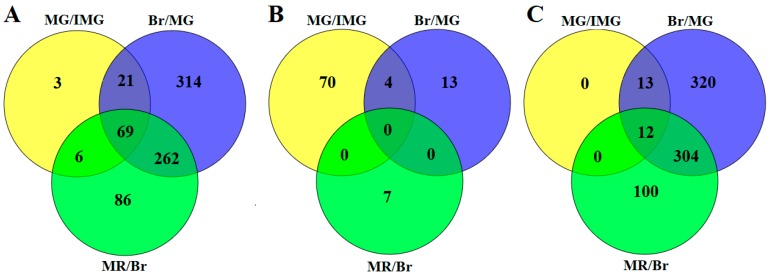
Overlap of the differentially expressed phosphoproteins (DEPPs) in Venn diagrams with different comparisons. (**A**) Total DEPPs; (**B**) upregulated DEPPs; (**C**) downregulated DEPPs.

**Figure 3 ijms-21-01962-f003:**
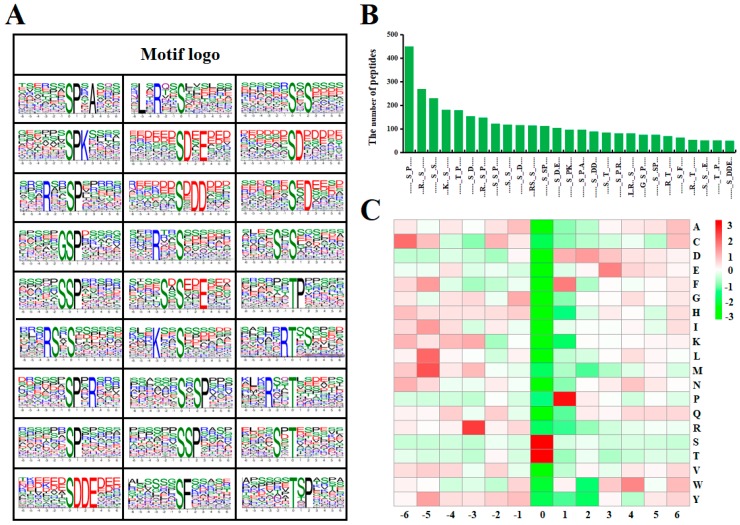
Analysis of phosphorylation sites. (**A**) Sequence motif analysis of phosphorylation sites. (**B**) The number of identified peptides containing phosphorylation sites in each motif. (**C**) The relative abundance of amino acid residues flanking the phosphorylation sites represented by an intensity map. The intensity map shows the relative abundance for six amino acids from the phosphorylation site. The colors in the intensity map represent the log10 of the ratio of frequencies (red shows enrichment, green shows depletion).

**Figure 4 ijms-21-01962-f004:**
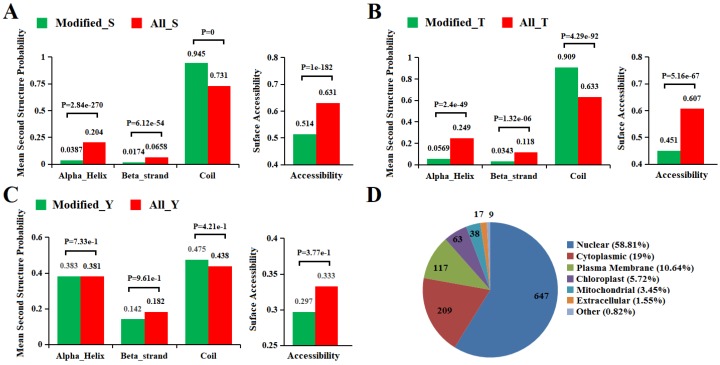
Bioinformational analysis of phosphorylation sites. (**A**–**C**) Comparison of phosphoproteins and nonphosphoprotein amino acids in protein secondary structures and surface accessibility. (**D**) Distribution of the DEPPs in subcellular compartments.

**Figure 5 ijms-21-01962-f005:**
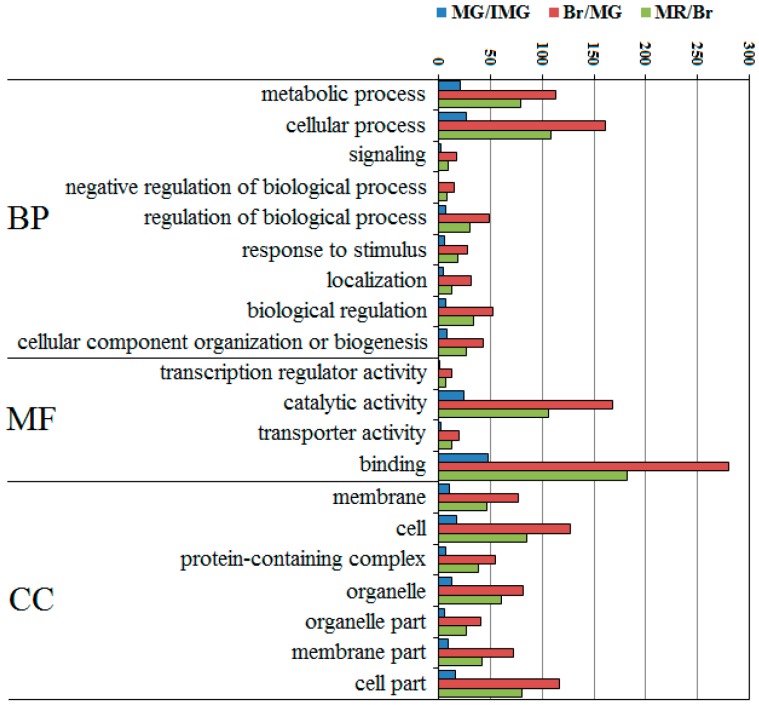
Top 20 GO categories assigned to the DEPPs. BP, biological process; MF, molecular function; CC, cellular component.

**Figure 6 ijms-21-01962-f006:**
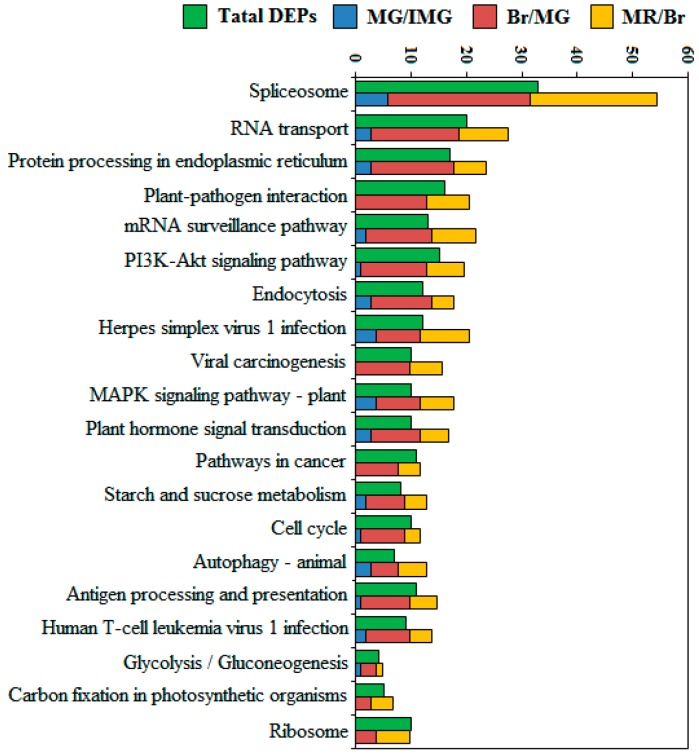
Top 20 KEGG pathways assigned to the DEPPs.

**Figure 7 ijms-21-01962-f007:**
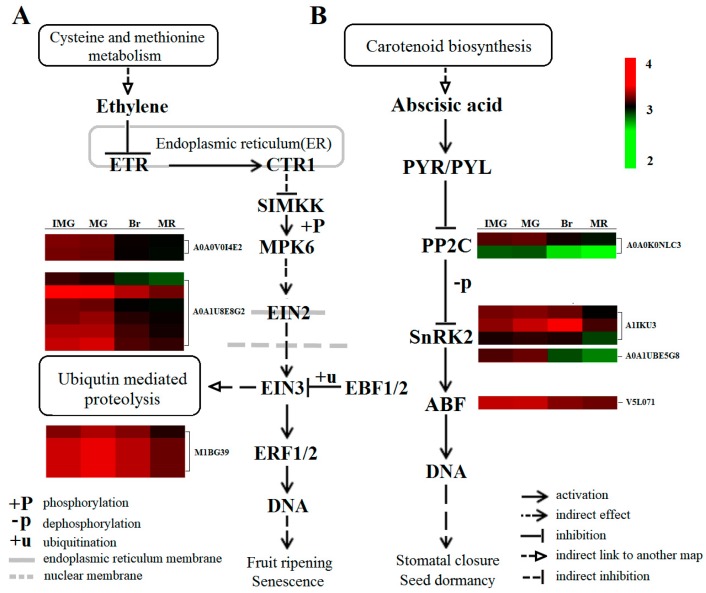
Comparative analysis of ethylene and the ABA signaling transduction pathway phosphoproteins from different development stages. IMG, immature green; MG, mature green; Br, breaker; MR, mature red. ETR, ethylene resistant; CTR1, constitutive triple response 1; SIMKK, mitogen-activated protein kinase kinase 4/5; MPK6, mitogen-activated protein kinase 6; EIN2, ethylene insensitive 2; EIN3, ethylene insensitive 3; EBF1/2, EIN3-binding F-box 1/2; ERF1/2, ethylene-responsive factors 1/2; PP2C, protein phosphatase 2C; SnRK2, SNF1-related protein kinase 2; ABF, ABA responsive element binding factor.

**Table 1 ijms-21-01962-t001:** Numbers of differentially expressed phosphoproteins in different comparisons.

Differential Expressed Proteins	MG/IMG	Br/MG	MR/Br
Upregulated	74	17	7
Downregulated	25	649	416
Total	99	666	423

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
