# Peer review of "Comprehensive Phosphoproteomic Analysis of Pepper Fruit Development Provides Insight into Plant Signaling Transduction"

_ijms, 2020, doi:10.3390/ijms21061962_

Round 1
Reviewer 1 Report
This study provides phosphoproteomic analysis of pepper fruit at four development stage by Tandem Mass Tag proteomic approaches. The study contains interesting elements and can be suitable for publication after some modifications.
L17: space and italic ‘pepper(Capsicum annuum L)’
L20-L29: Most of the abbreviations are not used again in the abstract.
L33: pepper
L54: In vivo – this expression is latin, should be in italic.
L61: 2-4%
L67: …18]. - space
L80: ..23] using
L86: al. [25]
L97: Please provide a clear objective statement.
L172: Lopato …
L200: [txP ]occupied
L206: leucine
L231: Phaeodactylum tricornutum – in italic
L241: [42].We – space
L251: 9,4 - space
L287: Figure should be self-explanatory. In the title, give also in full KEEG and DEPPs
L323: (Figure 7)
L403: rapidly.This - space
L408: Capsicum annuum – italic
L416: [71].The – space
L455: score.
L477: ).
L499: L.O.. – double points
L503: sequencing.
L511: Capsicum – in italic
L590: Brachypodium distachyon – italic
L610, 640, 653, 670: Arabidopsis thaliana – italic
L619: Oryza sativa
L630: TrendsPlant sci
Author Response
L17: space and italic ‘pepper(Capsicum annuum L)’
Response: Thank you for your comment. It has been modified as required.
L20-L29: Most of the abbreviations are not used again in the abstract.
Response:Thank you for your comment. It has been modified as required.
L33: pepper
Response:Thank you for your comment. It has been modified as required.
L54: In vivo – this expression is latin, should be in italic.
Response:Thank you for your comment. It has been modified as required.
L61: 2-4%
Response:Thank you for your comment. It has been modified as required.
L67: …18]. - space
Response:Thank you for your comment. It has been modified as required.
L80: ..23] using
ResponseThank you for your comment.: It has been modified as required.
L86: al. [25]
Response:Thank you for your comment. It has been modified as required.
L97: Please provide a clear objective statement.
Response:Thank you for your comment. A clear objective statement was provided in L89-L93.
L172: Lopato …
Response:Thank you for your comment. It has been modified as required.
L200: [txP ]occupied
Response:Thank you for your comment. It has been modified as required.
L206: leucine
Response:Thank you for your comment. It has been modified as required.
L231: Phaeodactylum tricornutum – in italic
Response:Thank you for your comment. It has been modified as required.
L241: [42].We – space
Response:Thank you for your comment. It has been modified as required.
L251: 9,4 - space
Response:Thank you for your comment. It has been modified as required.
L287: Figure should be self-explanatory. In the title, give also in full KEEG and DEPPs
Response:Thank you for your comment. The related title has been modified as you requested.
L323: (Figure 7)
Response:Thank you for your comment. It has been modified as required.
L403: rapidly.This - space
Response:Thank you for your comment. It has been modified as required.
L408: Capsicum annuum – italic
Response: Thank you for your comment. It has been modified as required.
L416: [71].The – space
Response: Thank you for your comment. It has been modified as required.
L455: score.
Response: Thank you for your comment. It has been modified as required.
L477: ).
Response: Thank you for your comment. It has been modified as required.
L499: L.O.. – double points
Response: Thank you for your comment. It has been modified as required.
L503: sequencing.
Response: Thank you for your comment. It has been modified as required.
L511: Capsicum – in italic
Response: Thank you for your comment. It has been modified as required.
L590: Brachypodium distachyon – italic
Response: Thank you for your comment. It has been modified as required.
L610, 640, 653, 670: Arabidopsis thaliana – italic
Response: Thank you for your comment. It has been modified as required.
L619: Oryza sativa
Response: Thank you for your comment. It has been modified as required.
L630: TrendsPlant sci
Response: Thank you for your comment. It has been modified as required.
Reviewer 2 Report
The manuscript of Liu et al. is in the field of the “International Journal of Molecular Science”, plant biology with proteomic experiment and measurement of phosphopeptides. The aim of this article is to analyse phosphopeptides in pepper fruit development and to make correlation with plant signal transduction.
This study is interesting but the discussion of the author’s results is poor concerning the different hypothesis about the role of phosphorylation of protein in fruit development. The same criticism can be made concerning the relationships between this post-translational modification and plant signal transduction. Moreover, there are many typographical errors (italics, missing spaces, ...) and also incorrect spelling.
Concerning the results of this article, I have a major criticism. I don’t understand how the authors identified the phosphorylated peptides by MS/MS. We have also no idea about the number of total identifications? Why the authors didn’t use the PRM mode to identify phosphorylated peptides like in Taumer et al, 2018 ?
This article can be accepted in “International Journal of Molecular Science”. But revisions must be brought in particular in the discussion of the role of phosphorylation in fruit development and in the identification of phophoprotein with MS/MS approach without identification with PRM mode. A great effort must also be made to rereading this article because there are still too many typographical errors and English can be improved.
Author Response
This study is interesting but the discussion of the author’s results is poor concerning the different hypothesis about the role of phosphorylation of protein in fruit development. The same criticism can be made concerning the relationships between this post-translational modification and plant signal transduction. Moreover, there are many typographical errors (italics, missing spaces, ...) and also incorrect spelling.
Response: Thank you for your advice. We have made appropriate additions and modifications about the role of phosphorylation of protein in fruit development and the relationships between this post-translational modification and plant signal transduction in accordance with your comments. However, at present there are few studies on large-scale phosphoproteome in fruit development, leading to few hypotheses on the role of protein phosphorylation in fruit development, which makes it impossible to fully discuss the results of this paper by referring to other hypotheses on phosphoproteome in fruit development. Therefore, on the basis of reference to other fruit development phosphorproteome, through the study of other plant development phosphoproteome, from the function of related phosphorylation protein to explore its role in pepper fruit development, in order to provide reference and basis for future research on protein phosphorylation in fruit development. And the typographical errors and incorrect spelling were carefully corrected.
Concerning the results of this article, I have a major criticism. I don’t understand how the authors identified the phosphorylated peptides by MS/MS. We have also no idea about the number of total identifications? Why the authors didn’t use the PRM mode to identify phosphorylated peptides like in Taumer et al, 2018 ?
Response: Thanks for pointing this out. For phosphorylated peptides identification during database matching, we set the variable modifications were phosphorylation modification for lysines, tyrosine and serine, now we have added the identification method in L463-L467 in the revised manuscript. And We have iddentified 4150 phosphosites, 2639 phosphopeptides and 1566 phosphoproteins and depicted in L98-L100. At present, the PRM verification technology is relatively new and is currently mainly used in proteomics studies. Currently, there are no articles indicating any credible phosphorylated peptides existing in the materials we studied. So, we need DDA phosphoproteomics to find phosphopeptides’ candidates. Taumer et al showed that reproducibility of PRM acquisition is superior to DDA in analysis of a relatively small number (< 100) of pre-selected target phosphopeptides, which fits well with the number of phosphopeptides detected in a typical bacterial phosphoproteomic LC-MS run. And they concluded that PRM shows a great promise for repetitive analyses of lowabundant bacterial phosphopeptides. An important reason they used PRM for bacterial phosphoproteomics is that the abundance of phosphorylated proteins in bacteria is much lower than in eukaryotes. So, considering that PRM verification technology application in phosphoproteome is relatively small, it costs more and lack of financial support, PRM verification has not been carried out in this study, but this method will be adopted in the follow-up in-depth study of phosphorylated proteins.
This article can be accepted in “International Journal of Molecular Science”. But revisions must be brought in particular in the discussion of the role of phosphorylation in fruit development and in the identification of phophoprotein with MS/MS approach without identification with PRM mode. A great effort must also be made to rereading this article because there are still too many typographical errors and English can be improved.
Response: Thank you for your comment. We have made appropriate additions and modifications in accordance with your comments, The detailed method of identifying phosphorylated proteins by MS/MS approach has been supplemented in manuscript. The typographical errors and english were carefully corrected.
Reviewer 3 Report
I have only few comments to text. The text of manuscript is sometimes confusing. The text has three subchapters named: Motif analysis of lysine phosporylated sites. Also there are mistakes in description of figures 1 and 3. The text should be revised again by authors.
Author Response
I have only few comments to text. The text of manuscript is sometimes confusing. The text has three subchapters named: Motif analysis of lysine phosporylated sites. Also there are mistakes in description of figures 1 and 3. The text should be revised again by authors.
Response: Thanks for pointing this out. We are very sorry that the three subchapters were named the same name due to our mistake, now these three subchapters have been amended as follows, subchapter 2.4 changed to Motif analysis of lysine phosphorylated peptides, subchapter 2.5 changed to Secondary structure and subcellular localization analysis, subchapter 2.5 changed to Plant hormone signal transduction in phosphorylated proteome. And We have modified the description mistakes in figures 1 and 3. And we have carefully modified the full text in the revised manuscript.